# Feasibility, Challenges, and Benefits of Global Antimicrobial Resistance Surveillance System Implementation: Results from a Multicenter Quasi-Experimental Study

**DOI:** 10.3390/antibiotics11030348

**Published:** 2022-03-06

**Authors:** Rujipas Sirijatuphat, Sunee Chayangsu, Jintana Srisompong, Darat Ruangkriengsin, Visanu Thamlikitkul, Surapee Tiengrim, Walaiporn Wangchinda, Pornpan Koomanachai, Pinyo Rattanaumpawan

**Affiliations:** 1Division of Infectious and Tropical Medicine, Department of Medicine, Faculty of Medicine Siriraj Hospital, Mahidol University, Bangkok 10700, Thailand; rujipas.sir@mahidol.ac.th (R.S.); visanu.tha@mahidol.ac.th (V.T.); surapee.tie@mahidol.ac.th (S.T.); walaiporn.wan@mahidol.ac.th (W.W.); nokmed@yahoo.com (P.K.); 2Department of Internal Medicine, Surin Hospital, Surin 32000, Thailand; chayangsu.sunee@gmail.com; 3Department of Internal Medicine, Suratthani Hospital, Suratthani 84000, Thailand; jint9839@gmail.com; 4Department of Internal Medicine, Sakaeo Crown Prince Hospital, Sakaeo 27000, Thailand; darat_r@yahoo.com

**Keywords:** antimicrobial resistance, antimicrobial stewardship, Global Antimicrobial Resistance Surveillance System (GLASS)

## Abstract

The Global Antimicrobial Resistance Surveillance System (GLASS) is one of the pillars of the global action plan on antimicrobial resistance launched by the World Health Organization in 2015. This study was conducted to determine the feasibility and benefits of GLASS as a component of antimicrobial stewardship strategies in three provincial hospitals in Thailand. Data on the types of bacteria isolated and their antibiotic susceptibility during January–December 2019 and January–April 2020 were retrieved from the microbiology laboratory of each participating hospital. Laboratory-based antibiograms from 2019 and GLASS-based antibiograms from 2020 were created and compared. A total of 14,877 and 3580 bacterial isolates were obtained during January–December 2019 and January–April 2020, respectively. The common bacteria isolated in both periods were *Escherichia coli*, *Klebsiella pneumoniae*, *Acinetobacter baumannii*, *Pseudomonas aeruginosa*, and *Staphylococcus aureus*. Hospital-acquired infection (HAI)-related bacteria were observed in 59.0%, whereas community-acquired infection (CAI)-related bacteria were observed in 41.0% of isolates. Antibiotic resistance in CAIs was high and may have been related to the misclassification of colonized bacteria as true pathogens and HAIs as CAIs. The results of this study on AMR surveillance using GLASS methodology may not be valid owing to several inadequate data collections and the problem of specimen contamination. Given these considerations, related personnel should receive additional training on the best practices in specimen collection and the management of AMR surveillance data using the GLASS approach.

## 1. Introduction

The burden of antimicrobial resistance (AMR) has been increasing worldwide, including in Thailand [1,2,3]. AMR surveillance is one of the pillars of the global action plan on AMR launched by the World Health Organization (WHO) in 2015. AMR surveillance (i) is an effective strategy to gather vital information to target the interventions for prevention and control of AMR; (ii) permits measurement of the outcomes of such interventions; (iii) enables early detection of AMR as well as early implementation of interventions; (iv) can reduce the burden of AMR [4].

In 2015, the WHO recommended an innovative surveillance system for AMR: the Global Antimicrobial Resistance Surveillance System (GLASS) [5,6]. GLASS is used for priority clinical specimens (i.e., blood, urine, feces, and genital samples) that are sent routinely for microbiological examinations for priority pathogens to test their susceptibility to particular antibiotics. GLASS is a case-based surveillance system that combines microbiological data with clinical and epidemiological data, such as the demographic features of patients, community-acquired infection (CAI), and hospital-acquired infection (HAI). A 2-day cutoff criterion is employed to differentiate between CAI and HAI. GLASS also de-duplicates repeated isolates from the same patient, but does not differentiate between true pathogens and colonized bacteria. Therefore, the bacterial isolates in a GLASS report include true pathogens and colonized bacteria. 

Studies on AMR surveillance in the bacteria recovered from blood, urine, and sputum samples were conducted at a large university hospital in Thailand. The authors revealed that the AMR surveillance data of clinical specimens using the GLASS methodology with differentiation between true pathogens and colonized bacteria, as well as differentiation between CAI and HAI using stringent definitions in addition to the 2-day cutoff criterion, were more valid and more applicable for monitoring AMR, and were more reliable for developing local guidelines for antibiotic treatment than the data obtained from a conventional laboratory-based surveillance system or conventional GLASS [7,8,9]. 

We undertook a quasi-experimental study to evaluate the impact of GLASS implementation in three provincial hospitals in Thailand. We aimed to determine the feasibility, challenges, and benefits of GLASS implementation in non-university hospitals in Thailand.

## 2. Materials and Methods

This was a quasi-experimental (pre- and post-implementation) study at three provincial hospitals in Thailand: Sakaeo Crown Prince Hospital (400 beds, Sakaeo Province, Eastern Thailand); Surin Hospital (900 beds, Surin Province, Northeastern Thailand); Surat Thani Hospital (800 beds, Surat Thani Province, Southern Thailand).

The study protocol was approved (384/2019) by the Ethics Review Board of the Faculty of Medicine Siriraj Hospital within Mahidol University (Bangkok, Thailand) and the ethics review boards of the three participating hospitals. Waiver of written informed consent from patients was granted because such antimicrobial stewardship activities are considered part of the quality of care improvement program. The study comprised three phases, as described below.

Phase 1: Pre-implementation phase (January–December 2019)

All culture specimens collected from patients from the three participating hospitals in 2019 were included. These specimens were sent to the microbiology laboratory. They were processed according to the standard operating procedures of the microbiology laboratory. The species of bacteria isolated and the antimicrobial susceptibility test (AST) results of the isolated bacteria were retrieved from the microbiology laboratory. A conventional laboratory-based antibiogram from 2019 was prepared and used as baseline data.

Phase 2: Implementation phase (end of December 2020)

An on-site visit and strategic planning meeting involving an experienced microbiologist from Mahidol University and a local microbiology team were held at each participating hospital to improve the capability and quality of microbiological study of clinical specimens and data management. 

Phase 3: Post-implementation phase (January–April 2020)

The isolated bacteria and AST results of priority specimens (blood, sputum, and urine) collected during January to April 2020 were retrieved from the microbiology laboratory. Because of the large number of isolated bacteria in each hospital, we randomly sampled approximately one-third of isolates recovered from blood, sputum, and urine from each hospital. The medical records of patients whose specimens were included were reviewed to determine if the isolated bacteria were true pathogens or colonized bacteria. The data of these selected patients were subsequently used to prepare antibiograms using the GLASS approach. 

### 2.1. Details of the GLASS Approach

The GLASS approach has three main components, as detailed below.

#### 2.1.1. Priority Pathogens and Specimen Types

Although GLASS focuses on blood, urine, stool, and genital samples, we included the specimens and priority pathogens shown below:

Blood: *Acinetobacter baumannii*, *Escherichia coli*, *Klebsiella pneumoniae*, *Pseudomonas aeruginosa*, and *Staphylococcus aureus*

Sputum: *A. baumannii*, *E. coli*, *K. pneumoniae*, *P. aeruginosa*, and *S. aureus*

Urine: *E. coli*, *K. pneumoniae*, *Enterococcus faecalis*, and *Enterococcus faecium*

#### 2.1.2. De-Duplication

If several cultures were collected during patient care, the duplicate isolated bacteria from the same patient were excluded and only the first isolate is reported for each patient per surveyed specimen type and tested pathogen.

#### 2.1.3. Origin of Infection

CAI was defined as an infection in a given patient who had been treated as an outpatient or had been admitted for ≤2 calendar days at the time of specimen collection. 

HAI was defined as an infection in a given patient who had been admitted for >2 calendar days at the time of specimen collection. 

Data from clinical specimens sent to the microbiology laboratory during January to April 2020 underwent additional management to ascertain if the isolates were true pathogens or colonized bacteria. This aim was achieved by reviewing patients’ medical records, and only true pathogens were included in the analyses. A colonized bacterial isolate was defined as a bacterium isolated from a given patient who had no clinical features of an infection or the patient recovered from a suspected infection without antimicrobial therapy. 

### 2.2. Collection and Analyses of Data

Conventional antibiograms from 2019 were analyzed using only the data obtained from the microbiology laboratory. The GLASS-based antibiograms from 2020 were analyzed using the data obtained from the microbiology laboratory and review of patients’ medical records. 

The coronavirus disease 2019 (COVID-19) outbreak accelerated in late April 2020 in Thailand; therefore, the study team decided to include only the bacteria isolated during January–April 2020. The estimated number of bacteria isolated was 1500 isolates per hospital during these 4 months of the post-implementation period. Therefore, the sample size was sufficient to determine the prevalence of antimicrobial resistance at 10–90% with an allowable error of 10%. 

The review of patients’ medical records was completed during the post-implementation period. The necessary data included the demographic characteristics of patients, nature of isolated bacteria (true infection or colonization), origin of infection (CAI or HAI) using the 2-day cutoff criterion, and risk of multidrug resistance (MDR). The latter was defined as the presence of at least one healthcare-associated condition, including: being admitted to another hospital for >2 days before transfer to the study hospital; hospitalization within the previous 3 months; antimicrobial therapy within the previous 3 months; being a resident at a long-term care facility; chronic hemodialysis.

After obtaining all necessary data, the 2020 GLASS-based antibiogram of each hospital was processed by WHONET 2020 (www.whonet.org/, accessed on 30 April 2021).

## 3. Results

### 3.1. Data during the Preimplementation Phase (January–December 2019)

A total of 14,877 bacterial isolates were recovered from all clinical specimens in the three participating hospitals. Among the 14,877 bacterial isolates, the common bacteria isolated were *K. pneumoniae* (2591 isolates, 17.4%)*, A. baumannii* (2270 isolates, 15.3%)*, E. coli* (2195 isolates, 14.8%), *P. aeruginosa* (1459 isolates, 9.8%), and *S. aureus* (608 isolates, 4.1%). The antimicrobial susceptibility of the aforementioned common isolates is shown in Table 1, Table 2, Table 3, Table 4 and Table 5.

### 3.2. Data during the Postimplementation Phase (January–April 2020)

A total of 3580 isolates from 3190 unique patients were included. Among the 3580 isolates, 3451 of them (96.4%) were determined to be causative isolates, whereas 129 of them (3.6%) were considered to be colonized isolates. The most common bacterium isolated was *E. coli* (930 isolates, 26.9%), followed by *K. pneumoniae* (800 isolates, 23.2%)*, A. baumannii* (482 isolates, 14.0%)*, P. aeruginosa* (396 isolates, 11.5%), and *S. aureus* (231 isolates, 6.7%). We found that 2041 isolates (59.1%) were HAI-related bacteria, whereas 1410 isolates (40.9%) were CAI-related bacteria. MDR-risk was identified in 443 isolates (31.4%) of CAI-related bacteria. 

Among the 1410 CAI-related bacterial isolates, the most common bacterium isolated was *E. coli* (596 isolates, 42.3%)*,* followed by *K. pneumoniae* (348 isolates, 24.7%), *S. aureus* (148 isolates, 10.5%), *P. aeruginosa* (96 isolates, 6.8%), and *A. baumannii* (69 isolates, 4.9%). Among the 2041 HAI-related bacterial isolates, the most common bacterium isolated was *K. pneumoniae* (452 isolates, 22.1%)*,* followed by *A. baumannii* (413 isolates, 20.2%), *E. coli* (334 isolates, 16.4%), *P. aeruginosa* (300 isolates, 14.7%), *E. faecalis* (130 isolates, 6.4%), *E. faecium* (89 isolates, 4.4%) and *S. aureus* (83 isolates, 4.1%). 

A total of 875 bacterial isolates were recovered from blood specimens. Of these blood-culture isolates, the most common bacterium isolated was *E. coli* (324 isolates, 37.0%)*,* followed by *K. pneumoniae* (198 isolates, 22.6%), *S. aureus* (120 isolates, 13.7%), *A. baumannii* (86 isolates, 9.8%), *Burkholderia pseudomallei* (53 isolates, 6.1%), and *P. aeruginosa* (38 isolates, 4.3%). A total of 1385 bacterial isolates were recovered from sputum specimens. Of these sputum-culture isolates, the most common bacterium isolated was *K. pneumoniae* (375 isolates, 27.1%), followed by *A. baumannii* (322 isolates, 23.2%), *P. aeruginosa* (288 isolates, 20.3%), *E. coli* (121 isolates, 8.7%), *S. aureus* (111 isolates, 8.0%), and *Enterobacter cloacae* (78 isolates, 5.6%). A total of 1,191 bacterial isolates were recovered from urine specimens. Of these urine-culture isolates, the most common bacterium isolated was *E. coli* (485 isolates, 40.7%), followed by *K. pneumoniae* (277 isolates, 23.3%), *E. faecalis* (160 isolates, 13.4%), *E. faecium* (128 isolates, 10.7%), *P. aeruginosa* (75 isolates, 6.3%), and *A. baumannii* (66 isolates, 5.5%). The antimicrobial susceptibility of the aforementioned common isolates is shown in Table 1, Table 2, Table 3, Table 4 and Table 5.

Comparison between the conventional antibiogram made from microbiology data in 2019 and the GLASS antibiogram made from microbiology data and some clinical data in 2020 revealed the prevalence of ceftriaxone resistance in *E. coli* isolates in 2019 and 2020 was 44% and 53%, respectively. The prevalence of ceftriaxone-resistant *K. pneumoniae* was 41% in 2019 and 44% in 2020. A higher prevalence of ceftazidime, piperacillin/tazobactam, and ciprofloxacin resistances for isolates of *E. coli*, *K. pneumoniae*, *A. baumannii*, and *P. aeruginosa* was observed in 2019 compared to 2020. The prevalence of meropenem-resistant *A. baumannii* and *P. aeruginosa* was 61% and 23%, respectively, in 2019, whereas it was 80% and 32%, respectively, in 2020. Methicillin-resistant *S. aureus* (MRSA) was found in 4% of *S. aureus* isolates from 2019 and in 9% of *S. aureus* isolates from 2020.

The resistance of *E. coli*, *K. pneumoniae*, *A. baumannii*, and *P. aeruginosa* isolated from blood specimens was less than that from sputum and urine specimens. The resistance profiles of *S. aureus* isolates from blood specimens were comparable to those of sputum specimens.

We compared the antibiotic susceptibility between CAI- and HAI-related isolates. Hospital-acquired *E. coli* and *K. pneumoniae* isolates were more resistant to ceftriaxone than community-acquired isolates. Most hospital-acquired *E. coli*, *K. pneumoniae*, *A. baumannii*, and *P. aeruginosa* isolates were more resistant to ceftazidime, ciprofloxacin, trimethoprim/sulfamethoxazole, piperacillin/tazobactam, and meropenem than community-acquired isolates. MRSA was observed in 4% of community-acquired *S. aureus* isolates from blood specimens, whereas 6% of hospital-acquired *S. aureus* isolates from blood specimens were MRSA.

The isolates recovered from clinical specimens in 2020 were classified as colonized bacteria in 3.6% of isolates. In addition, 31.4% of patients with positive cultures in 2020 who were classified as CAI according to the 2-day cutoff criterion had a risk of resistance to at least one antibiotic.

## 4. Discussion

The data of isolated bacteria and their AST results comprised all isolates in the whole of 2019, but included bacteria isolated during January to April 2020 because of the COVID-19 pandemic. Therefore, the data reported during January to April 2020 might differ from the data collected from January to December 2020. Therefore, direct comparison of the antimicrobial resistance of each bacterium between 2019 and 2020 may be inaccurate because bacterial isolates in 2019 were recovered from all specimens but those in 2020, which were recovered from only blood, urine, and sputum specimens. Moreover, the data for all isolates during January to April 2020 were compared with the data for only the isolates considered to be true pathogens in 2020. Therefore, comparing the data between these two periods might be challenging.

The distribution of bacterial isolates varied across the different types of clinical specimens and did not change after implementation of GLASS (Table 1, Table 2, Table 3, Table 4 and Table 5). The types of common bacteria isolated from blood, urine, and sputum specimens and the higher prevalence of resistance to most antimicrobial agents in the isolates from HAI than that from CAI observed in the present study are similar to those reported in studies conducted in other hospitals in Thailand [7,8,9]. 

Most of the types of isolates in 2020 seemed to be more resistant than the isolates in 2019 even though the isolates in 2020 were assumed to be true pathogens. This observation might be due to sampling variation in 2020, whereby the isolates were collected for only 4 months and/or because the patients in 2020 had more HAI and higher MDR risk than those in 2019.

The information on AMR surveillance of bacterial isolates recovered in clinical specimens in 2020 observed in the present study may not be appropriate for developing local guidelines for antibiotic use until two important issues related to the validity of such information are resolved. First, the best practices in specimen collection without specimen contamination should be adopted. This oversight was confirmed by the high prevalence of *E. coli* isolated from sputum specimens. Furthermore, the overall prevalence of colonized bacteria in this study (4%) is much lower than what was observed in previous studies (30–60%) [7,8,9]. It is reasonable to think that many isolates in this study were misclassified as true pathogens and were included in the AST results of 2020. Second, hospital personnel who retrieve information on MDR risk should improve their skill in classifying CAI. The data in Table 1, Table 2, Table 3, Table 4 and Table 5 show that the resistance of isolates in CAI classified by the 2-day cutoff criteria was high. For example (Figure 1), the prevalence of ceftriaxone-resistant *E. coli* isolated from urine in CAI was 53% when the time-based criterion was used and 46% when the revised criteria (time-based criterion with MDR risk) were used. This indicated that the 2-day cutoff criterion used to differentiate CAI and HAI is not sufficient and that MDR risk might be assessed inappropriately in all CAIs classified by the 2-day cutoff criterion. This phenomenon will have a huge impact on the recommendation of antibiotics for therapy of a suspected urinary tract infection (UTI) due to *E. coli* because the recommended first-line antibiotic is piperacillin/tazobactam, meropenem, or amikacin instead of ceftriaxone if the susceptibility of *E. coli* from urine specimens to ceftriaxone is only 47–53%. Usually, ceftriaxone is recommended as the first-line antimicrobial for empiric therapy of a community-acquired UTI because most isolates of *E. coli* causing a community-acquired UTI have remained susceptible to ceftriaxone in many studies [5]. Further evidence suggesting misclassification of HAI as CAI was that 4% of *S. aureus* isolates from blood and sputum specimens from CAI patients were MRSA. Community-associated MRSA is extremely rare in Thailand; therefore, these patients were more likely to have had HAI instead of CAI [10,11]. 

All participating hospitals in this study were assigned as the main healthcare facilities taking care of COVID-19 patients in their areas. Therefore, such a situation may have harmed the healthcare personnel’s physical and mental health [12,13]. Additionally, there may have been some delays in the study processes such as specimen collection, microbiology laboratory training, and data collection [14,15,16]. 

As mentioned above, the results of our study on AMR surveillance using the GLASS methodology in 2020 encountered several problems. Nevertheless, the data derived from GLASS with additional valid information should be useful for AMR monitoring and developing antibiotic guidelines for patients suspected of having bacterial infections (bacteremia, pneumonia, and UTIs). In addition, the personnel collecting samples should be trained thoroughly to overcome practical problems. However, the GLASS approach consumes more time and resources than a laboratory-based surveillance system and annual AMR surveillance using the GLASS approach may not be feasible in most hospitals. Therefore, GLASS may be implemented every 2 years because we do not expect to see dramatic changes in the AMR of the target bacteria within this time.

## 5. Conclusions

The GLASS approach for AMR surveillance is feasible but has some challenges. The information obtained from an appropriate GLASS approach should be more valid than that from a conventional laboratory-based surveillance approach in different settings (CAI and HAI) in terms of (i) aggregating the antimicrobial susceptibility data of causative bacteria isolated for each clinical specimen and (ii) developing local guidelines for antibiotic treatment for patients with specific bacterial infections (bacteremia, pneumonia, and UTIs). The results from the GLASS approach would be more reliable if the best practices in specimen collection are widely adopted. Furthermore, the relevant personnel should receive more training on managing AMR surveillance data using the GLASS approach to obtain valid and applicable local information for antibiotic therapy of infected patients. 

## Figures and Tables

**Figure 1 antibiotics-11-00348-f001:**
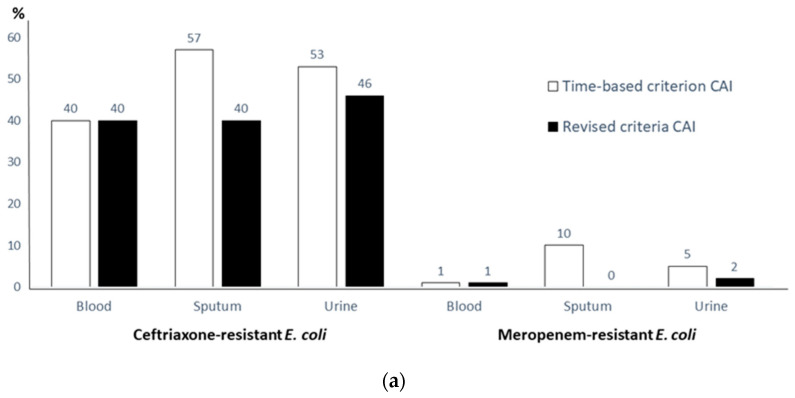
Prevalence of resistant bacteria when using GLASS-based CAI criteria (only time-based criterion) and revised criteria of CAI (time-based criterion with MDR risk). (**a**) Ceftriaxone-resistant *E. coli* and meropenem-resistant *E. coli*; (**b**) ceftriaxone-resistant *K. pneumoniae* and meropenem-resistant *K. pneumoniae*.

**Table 1 antibiotics-11-00348-t001:** Antibiotic susceptibility profiles of *E. coli* isolated in 2019 and 2020.

Specimens (Year)	Types (*n*)	Percentage of Susceptibility						
		Ceftriaxone	Ceftazidime	Pip/Taz	Meropenem	Ciprofloxacin	Amikacin	TMP/SMX
All (2019)	All (*n =* 2195)	56	69	91	93	50	94	45
All * (2020)	All (*n =* 793)	47	61	89	95	46	98	41
Blood	All (2019) (*n =* 659)	71	84	94	96	79	92	55
	All * (2020) (*n =* 314)	59	73	78	97	53	99	50
	CAI (*n =* 214)	60	75	74	99	60	99	62
	HAI (*n =* 100)	58	69	83	95	36	99	39
Sputum	All (2019) (*n =* 391)	48	62	87	90	74	87	44
	All * (2020) (*n =* 120)	41	57	97	94	53	97	42
	CAI (*n =* 65)	43	55	92	90	52	100	52
	HAI (*n =* 55)	39	60	100	100	56	92	33
Urine	All (2019) (*n =* 1237)	52	64	89	92	70	87	40
	All * (2020) (*n =* 458)	42	55	89	93	37	98	38
	CAI (*n =* 288)	47	59	92	95	37	100	41
	HAI (*n =* 170)	35	48	83	90	38	94	27

* Included only true pathogens. Abbreviations: *n,* number of isolates; Pip/taz, piperacillin/tazobactam; TMP/SMX, trimethoprim/sulfamethoxazole; CAI, community-acquired infection; HAI, hospital-acquired infection.

**Table 2 antibiotics-11-00348-t002:** Antibiotic susceptibility profiles of *K. pneumoniae* isolated in 2019 and 2020.

Specimens (Year)	Types (*n*)	Percentage of Susceptibility						
		Ceftriaxone	Ceftazidime	Pip/Taz	Meropenem	Ciprofloxacin	Amikacin	TMP/SMX
All (2019)	All (*n =* 2591)	59	65	75	88	62	92	63
All * (2020)	All (*n =* 684)	56	55	70	89	56	94	59
Blood	All (2019) (*n =* 307)	63	70	67	86	88	90	61
	All * (2020) (*n =* 179)	70	64	66	91	72	96	52
	CAI (*n =* 101)	85	80	85	98	83	99	65
	HAI (*n =* 78)	50	46	53	82	53	94	43
Sputum	All (2019) (*n =* 1864)	59	66	78	90	87	89	65
	All * (2020) (*n =* 359)	57	57	73	93	65	99	63
	CAI (*n =* 143)	73	75	82	97	81	99	73
	HAI (*n =* 55)	48	46	67	91	54	98	58
Urine	All (2019) (*n =* 480)	47	52	66	81	72	81	51
	All * (2020) (*n =* 222)	41	39	62	82	28	86	59
	CAI (*n =* 95)	63	54	62	90	34	94	61
	HAI (*n =* 127)	28	27	63	76	26	80	55

* Included only true pathogens. Abbreviations: *n,* number of isolates; Pip/taz, piperacillin/tazobactam; TMP/SMX, trimethoprim/sulfamethoxazole; CAI, community-acquired infection; HAI, hospital-acquired infection.

**Table 3 antibiotics-11-00348-t003:** Antibiotic susceptibility profiles of *A. baumannii* isolated in 2019 and 2020.

Specimens (Year)	Types (*n*)	Percentage of Susceptibility						
		Ceftriaxone	Ceftazidime	Pip/Taz	Meropenem	Ciprofloxacin	Amikacin	TMP/SMX
All (2019)	All (*n =* 2270)	17	38	35	39	40	60	49
All * (2020)	All (*n =* 429)	7	16	25	20	15	54	36
Blood	All (2019) (*n =* 240)	33	51	50	55	58	75	52
	All * (2020) (*n =* 88)	27	33	41	44	46	73	47
	CAI (*n =* 20)	31	86	75	93	100	93	71
	HAI (*n =* 68)	11	20	31	32	35	67	40
Sputum	All (2019) (*n =* 1894)	7	35	32	37	43	57	47
	All * (2020) (*n =* 303)	3	12	21	13	5	45	33
	CAI (*n =* 34)	8	21	23	28	33	57	32
	HAI (*n =* 269)	2	12	21	11	4	44	34
Urine	All (2019) (*n =* 210)	8	29	31	28	33	63	44
	All * (2020) (*n =* 73)	3	12	20	20	13	64	25
	CAI (*n =* 12)	0	0	-	0	0	67	-
	HAI (*n =* 61)	4	13	20	21	14	64	25

* Included only true pathogens. Abbreviations: *n,* number of isolates; Pip/taz, piperacillin/tazobactam; TMP/SMX, trimethoprim/sulfamethoxazole; CAI, community-acquired infection; HAI, hospital-acquired infection.

**Table 4 antibiotics-11-00348-t004:** Antibiotic susceptibility profiles of *P. aeruginosa* isolated in 2019 and 2020.

Specimens (Year)	Types (*n*)	Percentage of Susceptibility				
		Ceftazidime	Pip/Taz	Meropenem	Ciprofloxacin	Amikacin
All (2019)	All (*n =* 1459)	74	81	77	80	90
All * (2020)	All (*n =* 368)	65	72	68	69	80
Blood	All (2019) (*n =* 90)	75	87	70	59	93
	All * (2020) (*n =* 38)	74	73	58	78	81
	CAI (*n =* 11)	100	100	91	100	100
	HAI (*n =* 27)	63	57	44	72	73
Sputum	All (2019) (*n =* 1061)	80	85	82	83	93
	All * (2020) (*n =* 275)	72	77	76	83	91
	CAI (*n =* 67)	84	79	81	91	95
	HAI (*n =* 208)	69	77	74	81	89
Urine	All (2019) (*n =* 294)	47	55	53	42	57
	All * (2020) (*n =* 75)	34	46	43	29	36
	CAI (*n =* 17)	38	50	38	25	25
	HAI (*n =* 58)	33	44	43	32	38

* Included only true pathogens. Abbreviations: *n,* number of isolates; Pip/taz, piperacillin/tazobactam; TMP/SMX, trimethoprim/sulfamethoxazole; CAI, community-acquired infection; HAI, hospital-acquired infection.

**Table 5 antibiotics-11-00348-t005:** Antibiotic susceptibility profiles of *S. aureus* isolated in 2019 and 2020.

Specimens (Year)	Types (*n*)	Percentage of Susceptibility					
		Oxacillin	Ciprofloxacin	Clindamycin	Erythromycin	TMP/SMX	Vancomycin
All (2019)	All (*n =* 608)	91	89	89	91	95	100
All * (2020)	All (*n =* 221)	96	98	87	86	95	100
Blood	All (2019) (*n =* 211)	86	92	87	90	92	100
	All * (2020) (*n =* 118)	96	98	89	88	90	100
	CAI (*n =* 74)	96	97	91	90	89	100
	HAI (*n =* 44)	94	100	86	86	93	100
Sputum	All (2019) (*n =* 194)	97	95	89	94	97	100
	All * (2020) (*n =* 110)	97	97	85	85	98	100
	CAI (*n =* 72)	96	98	88	88	97	100
	HAI (*n =* 38)	100	95	81	81	100	100
Urine **	-	-	-	-	-	-	-
	-	-	-	-	-	-	-
	-	-	-	-	-	-	-
	-	-	-	-	-	-	-

* Included only true pathogens. Abbreviations: *n,* number of isolates; TMP/SMX, trimethoprim/sulfamethoxazole; CAI, community-acquired infection; HAI, hospital-acquired infection. ** *S. aureus* was not isolated from urine cultures.

## Data Availability

The study dataset is available from the corresponding author upon reasonable request.

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
