# Peer review of "Feasibility, Challenges, and Benefits of Global Antimicrobial Resistance Surveillance System Implementation: Results from a Multicenter Quasi-Experimental Study"

_antibiotics, 2022, doi:10.3390/antibiotics11030348_

Round 1
Reviewer 1 Report
In this paper titled “Implementation of Global Antimicrobial Resistance Surveil- 2 lance System: Results from a Multicenter Quasi-experimental 3 Study” the authors conducted a GLASS study to see if GLASS might be used as part of antimicrobial stewardship strategies and reported the results. Because of multiple insufficient data collections, the results of this study on AMR surveillance utilizing GLASS approach may not be reliable. But, Data produced using GLASS, however, should be valuable for AMR monitoring and generating antibiotic treatment guidelines as more valid information. Overall, the article presents research that is of interest to researchers in the field of antibiotics research and that is well designed and performed with an appropriate experimental date.
Author Response
Please see the attachment (page 2).

Reviewer 2 Report
In the face of still rising antibiotic resistance especially during COVID 19 pandemic when disinfectants are widely used and could incude additionally cross-resistance, presented study is very valuable and crucial in actual microbiology.
My comments:
In the results section the detiled number of particular species of isolated bacteria should be presented and their precentage share in CAI and HAI isolates. Maybe a graph would be suitable.
I am also wondering that a shortened description of applied by labolatory microbiological methods should be included.
Author Response
Please see the attachment - page 3.

Reviewer 3 Report
Dear Editor,
MDPI-Antibiotics
I have evaluated the manuscript (Antibiotics-1613194) titled “Implementation of Global Antimicrobial Resistance Surveillance System: Results from a Multicenter Quasi-experimental Study” by Rattanaumpawan and co-authors. The authors have discussed the feasibility and benefit of GLASS as a component of antimicrobial stewardship strategies in three provincial hospitals in Thailand and which would be useful for local guidance for antibiotic treatment.
I am delighted to review this manuscript, covering an important aspect of the subject, with a good presentation of results, however, the author could have put more effort towards the discussion of the results. However, the topic of this manuscript makes this article interested to the readers of the journal Antibiotics. The manuscript follows the scope of the journal Antibiotics.
I would recommend the article could be published in Antibiotics with minor corrections. And the authors need to address the below-mentioned queries.
1. All the data were taken almost two years ago, any reason why the authors took so long to prepare the manuscript.
2. The author could re-check the timeline mentioned in phases 1, 2, and 3 of Materials and Methods and Discussion.
3. The author could show the three main components GLASS approach in the table.
4. Line 140”: Change “20202” to 2020.
5. Footnote is needed for missing urine data in table 5.
6. The author could have discussed results elaborately as the main pillar of this manuscript is based on the discussion of results as the author failed to collect all data due to Covid-19.
7. The author could include the following relevant references:
(a) Majumder, M., Rahman, S., Cohall, D., Bharatha, A., Singh, K., Haque, M., & Gittens-St Hilaire, M. (2020). Antimicrobial Stewardship: Fighting Antimicrobial Resistance and Protecting Global Public Health. Infection and drug resistance, 13, 4713–4738. https://doi.org/10.2147/IDR.S290835
(b) Valencia, C., Hammami, N., Agodi, A., Lepape, A., Herrejon, E. P., Blot, S., Vincent, J. L., & Lambert, M. L. (2016). Poor adherence to guidelines for preventing central line-associated bloodstream infections (CLABSI): results of a worldwide survey. Antimicrobial resistance and infection control, 5, 49. https://doi.org/10.1186/s13756-016-0139-y
Author Response
Please see the attachment - page 4-5.
